# Rapid Simultaneous Determination of 11 Synthetic Cannabinoids in Urine by Liquid Chromatography-Triple Quadrupole Mass Spectrometry

Jianbing Wu [1], Fan Zhang [2], Xing Ke [1], Wei Jia [2], Xuzhi Wan [2], Lange Zhang [2], Yilei Fan [1] and Jing Zhou [1,*]

[1] Key Laboratory of Drug Prevention and Control Technology of Zhejiang Province, Zhejiang Police College, Hangzhou 310053, China

[2] Key Laboratory of Agro-Products Postharvest Handling of Ministry of Agriculture and Rural Affairs, Zhejiang Key Laboratory for Agro-Food Processing, College of Biosystems Engineering and Food Science, Zhejiang University, Hangzhou 310058, China

\* Correspondence: zhoujing@zjjcxy.cn

**Abstract:** Synthetic cannabinoids are a series of synthetic substances that mimic the effects of natural cannabinoids and produce a much stronger toxicity than natural cannabinoids, and they have become the most abused family of new psychoactive substances. A solid-phase extraction–liquid chromatography-triple quadrupole/linear ion trap mass spectrometry method has been developed to determine 11 synthetic cannabinoids in rat urine. Oasis HLB cartridge was selected to simultaneously extract synthetic cannabinoids for pretreatment. The effects of the loading solution and elution reagent volume on the recovery were investigated. The optimized acetonitrile proportion and elution reagent volume were determined by both high recovery and low solvent consumption. The results showed that the linear coefficients of determination of 11 types of synthetic cannabinoids ranged from 0.993 to 0.999, the limit of quantitation ranged from 0.01 to 0.1 ng/mL, and the spiked recoveries ranged from 69.90% to 118.39%. The research presented here provides a validated liquid chromatography tandem mass spectrometry method to accurately identify and quantitate synthetic cannabinoid metabolites in urine samples.

**Keywords:** synthetic cannabinoids; LC-MS/MS; simultaneous determination; quantitative analysis; urine





## 1. Introduction

Cannabinoids can be divided into natural cannabinoids, endogenous cannabinoids, and synthetic cannabinoids based on the sources. Synthetic cannabinoids (SCs) refer to the agonists of synthetic endogenous cannabinoid receptors (CB1 and CB2), which combine with cannabinoid receptors and have more potent physiological and pharmacological activities than natural cannabinoids [1]. SCs are mostly non-polar compounds with lipophilic properties composed of 22 to 26 carbon atoms [2]. SCs belong to the group of psychoactive substances, which includes benzoyl indoles, naphthyl indoles, cyclohexyl phenols, naphthyl pyrrole, and diamond indoles [3]. Aminoalkylindole JWH-018 is the first identified psychoactive ingredient from SC products [4]. Synthetic cannabinoids develop from naphthoyl indole (JWH-018) to formylazole (THJ), indazole (AKB-48), and then indolecarbonamide (MDMB-CHMINACA) [5].

Synthetic cannabinoids have many commercial names, such as Spice, Black Mamba, and Cold K2 [6]. They have emerged in many countries and regions of the world, especially among young people. SCs have been the most widely monitored new psychoactive substances in Europe since 2008 [3]. The increase in consumption of SCs is particularly significant compared with other drugs on the worldwide market. Small changes in the original SC structures will lead to the formation of new SCs, and thus many illegal manufacturers can easily produce new SCs. It is important to note that new SCs continue to emerge rapidly, presenting a constant challenge for detection and analysis.

Studies on the identification and toxicology of SCs have been carried out in recent years. SCs have been reported to have specific therapeutic effects. Ellert-Miklaszewska et al. found that SCs could activate the killing pathway of human glioma cells and had antitumor activity [7]. However, more negative effects were mentioned. Ingestion of cannabinoids could cause many diseases, as described in the literature, including psychosis [8], intoxication [9], tachycardia [10], changes in blood pressure [11], and atrial fibrillation [12]. Some reports declared fatality directly related to the toxic effects of synthetic cannabinoids [13,14]. Tomiyama et al. reported that SCs induced cell apoptosis by regulating the cascade of cystatin, stressing that the abuse of cannabinoids could lead to neurological brain problems [15]. Radaelli et al. concluded that SCs could activate CB1 receptors in cardiomyocytes, participate in the production of reactive oxygen species, ATP consumption, and cell death, and then induce cardiotoxicity [16]. These compounds are also being manufactured by clandestine laboratories for the sole purpose of illicit consumption and pose a significant challenge to the law enforcement, medical, and forensic toxicology communities.

Due to new SCs constantly emerging in the market, there is an urgent need to update the method for the simultaneous detection of SCs. The content of the primary SCs is very low after being metabolized in the body, and the matrix composition of biological samples is complex. Therefore, it is necessary to develop a method with a low limit of quantitation and high precision to detect trace synthetic cannabinoids in biological samples. Usually, blood and urine are the main biological samples tested for cannabinoids, while saliva and hair can also be used as critical biological samples for testing. The advantage of urine for testing drugs is that the sample pretreatment process is simple, and the collection process is non-invasive. There are many types of SCs, and the structures of different types of them vary greatly, which brings certain difficulties to the detection of SCs. The recently used methods for the detection and analysis of SCs mainly include thin-layer chromatography (TIC) [17], nuclear magnetic resonance (NMR) [18], infrared spectroscopy (IR) [19], gas chromatography–mass spectrometry (GC-MS) [20], liquid chromatography–mass spectrometry (LC-MS) [21], and supercritical fluid chromatography (SFC) [22]. Laith et al. developed GC-MS to identify and quantify three common SCs seized on the market, namely AB-FUBINACA, AB-CHMINACA, and XLR-11, with LODs (limit of detection) ranging from 0.15 to 17.89 μg/mL [23]. Dong et al. established LC-MS/MS for the identification and quantification of three major phytocannabinoids and their metabolites and four synthetic cannabinoids in human urine, with LODs ranging from 0.01 to 0.5 ng/mL, an extraction recovery rate of 50%, and a matrix effect of between 59.4% and 100.1% [23]. However, most of the available studies showed that the LODs were relatively high and the quantitative performance remained to be improved. LC-MS is primarily employed for the analysis of compounds with high thermal instability, strong polarity, lipophilicity, or high molecular weight, with the advantages of strong separation ability, low detection limit, high sensitivity, and high accuracy. Therefore, LC-MS is widely used to analyze multiple SCs in biological samples simultaneously. Solid-phase extraction is a critical step and contributes significantly to the final performance of the analytical method.

For our study, we selected 11 highly potent synthetic cannabinoids that may lead to many adverse reactions, including both conventional and novel ones. The selected compounds are presented in the supplementary materials. N-(1-Amino-3-methyl-1-oxobutan-2-yl)-1-pentyl-1H-indazole-3-carboxamide (AB-PINACA), N-(1-Amino-3-methyl-1-oxobutan-2-yl)-1-(cyclohexylmethyl)-1H-indazole-3-carboxamide (AB-CHMINACA), and N-(1-Methoxy-3,3-dimethyl-1-oxobutan-2-yl)-1-(4-fluorobenzyl)-1H-indazole-3-carboxamide (MDMB-FUBINACA) share a similar structure, containing N-alkyl imine and 1H-indole moieties. Their common feature is their extremely potent pharmacological effects and toxicity, which led to their classification as controlled substances. (1-Pentyl-1H-indol-3-yl)(4-methylnaphthalen-1-yl)methanone (JWH-122), 1-Pentyl-3-(4-methoxybenzoyl)indole (RCS-4), and methyl 2-[1-(5-fluoropentyl)-1H-indazole-3-carboxamido]-3,3-dimethylbutanoate (5F-ADB) also have similar structures, containing 1H-indole and arylketone moieties. These compounds are highly toxic and classified as controlled substances as well. ADB-FUBINACA has

a structure similar to that of AB-PINACA, AB-CHMINACA, and MDMB-FUBINACA, containing N-alkyl imine and 1H-indole moieties, but with even greater toxicity. 5F-MDMB-PICA, 5F-AMB, UR-144, and AMB-FUBINACA have similar structures, containing indole moieties and alkyl imine, but have different substituent groups that result in different pharmacological effects and toxicity. Overall, synthetic cannabinoids are characterized by their structural diversity, high toxicity, and complex pharmacological effects. Due to their potential for abuse and danger, effective monitoring and control of these substances are necessary.

This article presents an improved and fully validated LC-MS/MS method for simultaneously detecting 11 synthetic cannabinoids in rat urine. The method validation process involves the assessment of linearity, precision, accuracy, and limits of quantification. In addition, SC concentrations in urine were measured at different time points after the injection of synthetic cannabinoids in rats. This method provides a useful tool for the identification of SCs in rat urine and a reference for the assessment of SC internal biological exposure.

## 2. Materials and Methods

### 2.1. Chemicals and Reagents

Synthetic cannabinoid standard: N-(1-Amino-3-methyl-1-oxobutan-2-yl)-1-pe-ntyl-1H-indazole-3-carboxamide (AB-PINACA), (1-Pentyl-1H-indol-3-yl)(4-methylnaphthalen-1-yl)methanone (JWH-122), N-(1-Amino-3-methyl-1-oxobutan-2-yl)-1-(cyclohexylmethyl)-1H-indazole-3-carboxamide (AB-CHMINACA), 1-Pentyl-3-(4-methoxybenzoyl)indole (RCS-4), N-(1-Methoxy-3,3-dimethyl-1-oxobutan-2-yl)-1-(4-fluorobenzyl)-1H-indazole-3-carboxamide (MDMB-FUBINACA), N-[1-(amino-3,3-dimethyl-1-oxobutan-2-yl)-1-(4-fluorobenzyl)-1H-indazole-3-carboxamide (ADB-FUBINACA), methyl 2-[1-(5-fluoropentyl)-1H-indazole-3-carboxamido]-3,3-dimethylbutanoate (5F-ADB), Methyl-2-[[1-(5-fluoropentyl)indole-3-carbonyl]amino]-3,3-dimethyl-butanoate) (5F-MDMB-PICA), N-[[1-(5-Fluoropentyl)-1H-indazol-3-yl]carbonyl]valine methyl ester (5F-AMB), (1-Pentyl-1H-indol-3-yl)(2,2,3,3-tetramethylcyclopropyl)methanone (UR-144), and methyl (S)-2-[1-(4-fluorobenzyl)-1H-indazole-3-carboxamido]-3-methylbutanoate (AMB-FUBINACA) were obtained from Shanghai Institute of Criminal Science and Technology. Formic acid ($\geq$96%, HPLC grade) was purchased from ROE Scientific Inc. (Barksdale Professional Center, Newark, NJ, USA); methanol and acetonitrile (both HPLC-grade) were obtained from Merck (New York, NY, USA). Ultrapure water (Millipore, Bedford, MA, USA) was used throughout the whole study.

### 2.2. Animal Study

Blank urine of male rats was obtained from the Zhejiang Academy of Medical Sciences (Zhejiang, China). Male Sprague Dawley rats (n = 12, six weeks, weighing 190–210 g) were obtained from the Zhejiang Academy of Medical Sciences (Zhejiang, China) and acclimatized for one week before collection of urine in the experimental room (temperature $23 \pm 2$ °C, humidity $55 \pm 10$%, 12 h light–dark cycle). After a 1-week adaptation, all the rats were transferred to individual metabolic cages to collect the urine samples, which were stored at $-20$ °C for further study. All experimental procedures were approved by the Zhejiang Experimental Animal Center Experimental Animal Welfare and Ethical Review Committee (Approval ID: ZJCLA-IACUC-20100005; Hangzhou, China).

### 2.3. Working Standards and Calibration Standards

Stock standard solutions of 11 synthetic cannabinoids were prepared by dissolving 1 mg of analytes in 1 mL of acetonitrile to obtain the concentration of 1 mg/mL and were kept at $-20$ °C. Mixed stock standard solutions were prepared by diluting the stock standard solutions of 11 synthetic cannabinoids with acetonitrile to obtain a concentration of 10 μg/mL. Mixed working standard solutions were prepared by diluting mixed stock standard solutions with acetonitrile to obtain a series of concentrations according to the experiment's needs.

### 2.4. Pretreatment of Urine Samples

The synthetic cannabinoids in the urine sample were extracted using an Oasis HLB cartridge (3 cc 60 mg, Waters, St. Quentin en Yvelines, France). Four hundred microliters of urine sample was mixed with 600 μL of acetonitrile and 1 mL of ultrapure water and vortexed for 30 s. The SPE cartridge was preconditioned with 3 mL of methanol and then equilibrated with 6 mL of ultrapure water before the mixed solution was loaded onto the cartridge. After loading, 3 mL of methanol–water (5%, *v/v*) was added for washing, and 4 mL of methanol was added for elution. The eluant was dried under a nitrogen stream and then reconstituted with 200 μL of acetonitrile for analysis by LC-MS/MS.

### 2.5. Determination of Synthetic Cannabinoids by LC-MS/MS

An AB SCIEX ExionLCTM0AD XR high-performance liquid chromatography system coupled with an AB SCIEX QTRAP 6500+ triple quadrupole/linear ion trap mass spectrometer system (AB SCIEX, Framingham, MA, USA) was used for analysis. The analytes were separated on a Waters UPLC HSS T3 column (1.8 μm, 2.1 mm × 150 mm) with the column temperature set at 40 °C. The binary solvent system consisted of 0.1% (*v/v*) aqueous formic acid solution (A) and acetonitrile (B). The gradient profile of the binary pump was maintained at 90% A from 0 to 1.0 min, decreased to 0% A from 1.0 to 7.0 min, was maintained at 0% A from 7.0 to 11.0 min, went back to 90% A at 11.1 min, and finally was held for 5 min. The flow rate was 0.3 mL/min. The equipment was operated in electrospray positive ionization mode (ESI+) under the multiple reaction monitoring mode (MRM). Optimized source parameters were set as follows: curtain gas 35 Psi, collision gas medium, ion spray voltage 5500 V (ESI), temperature 550 °C, ion source gas 1 50 Psi, ion source gas 2 50 Psi. The optimized mass spectra data of 11 SCs are shown in Table 1, and the chromatographic mass spectra of the standard solution (10 ng/mL) are shown in Figure 1.

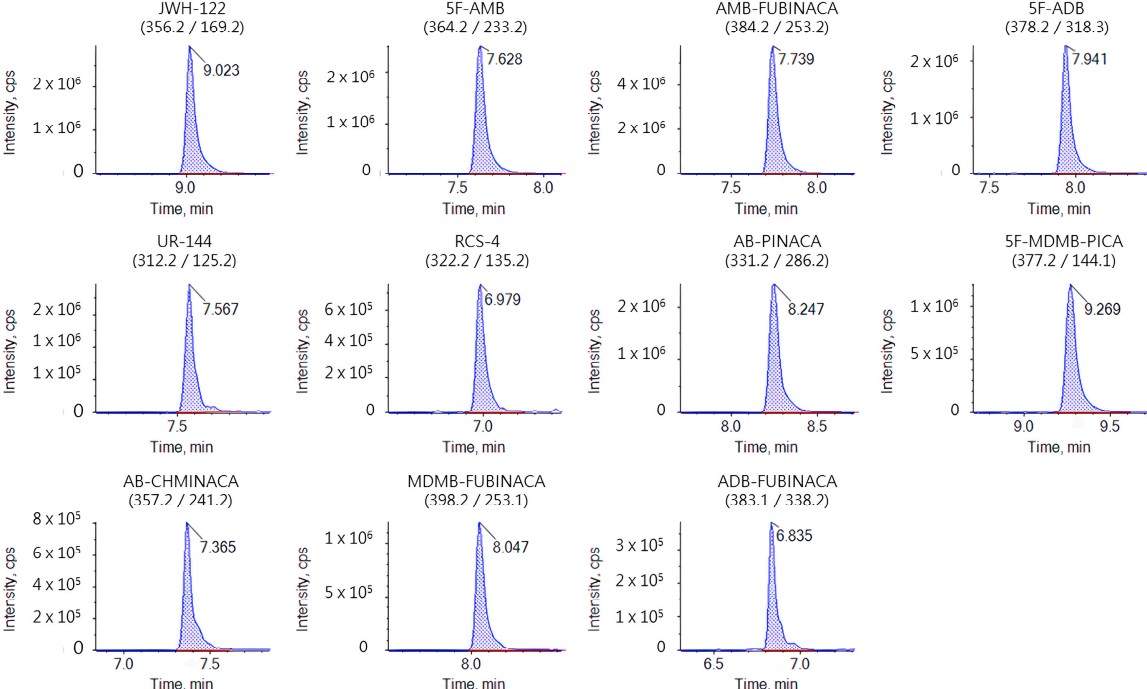

**Figure 1.** Chromatography–mass spectrometry of standard solution (10 ng/mL).

**Table 1.** Synthetic cannabinoid chromatography–mass spectrometry parameters.

| Compound | Molecular Formula | Retention Time (min) | Precursor Ion (*m/z*) | Daughter Ion (*m/z*) | Collision Energy (V) |
|---|---|---|---|---|---|
| JWH-122 | $C_{25}H_{25}NO$ | 8.98 | 356.2 | 169.2 [a]/214.2 | 32/32 |
| 5F-AMB | $C_{19}H_{26}FN_3O_3$ | 7.60 | 364.2 | 233.2 [a]/304.2 | 32/21 |
| AMB-FUBINACA | $C_{21}H_{21}FN_3O_3$ | 7.73 | 384.2 | 253.2 [a]/324.2 | 30/23 |
| UR-144 | $C_{21}H_{29}NO$ | 9.21 | 312.2 | 125.2 [a]/214.1 | 30/32 |
| RCS-4 | $C_{21}H_{23}NO_2$ | 8.21 | 322.2 | 135.2 [a]/214.2 | 31/32 |
| AB-CHMINACA | $C_{20}H_{28}N_4O_2$ | 7.34 | 357.2 | 241.2 [a]/169.2 | 34/35 |
| AB-PINACA | $C_{18}H_{26}N_4O_2$ | 6.95 | 331.2 | 286.2 [a]/314.2 | 21/13 |
| MDMB-FUBINACA | $C_{22}H_{24}FN_3O_3$ | 8.02 | 398.2 | 253.1 [a]/338.1 | 33/19 |
| 5F-ADB | $C_{20}H_{28}FN_3O_3$ | 7.91 | 378.2 | 223.1 [a]/318.3 | 31/23 |
| ADB-FUBINACA | $C_{21}H_{23}FN_4O_2$ | 6.82 | 383.1 | 253.1 [a]/338.2 | 34/23 |
| 5F-MDMB-PICA | $C_{21}H_{29}FN_2O_3$ | 7.53 | 377.2 | 144.1 [a]/232.2 | 56/20 |

[a] Quantifier ions.

*2.6. Method Validation*

The method's calibration curves, linear ranges, matrix effect, recovery, precision, and limit of quantification (LOQ) were determined according to Taverniers et al. for a robust method validation [24]. In simple terms, 11 synthetic cannabinoid standard compounds were added to pretreated blank rat urine to configure standard matrix curves. Blank rat urine samples were treated with the solid-phase extraction method, and then mixed standard substances of 11 SCs were added to prepare 0.1 ng/mL, 1.0 ng/mL, and 10.0 ng/mL spiked rat urine sample solution to calculate the matrix effect. We used intraday relative standard deviation to verify the precision of LC-MS/MS detection of SCs. The samples were injected six times at different times in a day, the average was taken, and the test results (n = 6) were analyzed. The recovery rate of this method examined the entire urine sample pretreatment step and was determined by the standard addition method. Recovery tests were performed by adding low, medium, and high levels of 11 standard solutions of cannabinoids to a blank matrix. The signal-to-noise ratio (S/N) of the blank matrix diluted to the HPLC MS/MS system was 10, and the limits of quantification were determined.

**3. Results and Discussion**

*3.1. Optimization of Sample Pretreatment*

For the pretreatment of urine matrix, liquid–liquid extraction or solid-phase extraction methods are usually adopted. As a conventional extraction method, liquid–liquid extraction has the disadvantages of large consumption of organic solvents and unstable recovery [25]. Solid-phase extraction, which is widely used, has the advantages of a high enrichment ratio, high accuracy, and clean matrix treatment compared with liquid–liquid extraction [26]. Yeter et al. used liquid chromatography–high-resolution mass spectrometry to determine 5F-ADB and its methyl ester metabolite in blood and urine samples, which were pretreated with a solid-phase extraction cartridge (OASIS HLB, 3 cc, 60 mg). The method was successfully applied to 70 human blood and 36 urine samples [27]. Our laboratory has investigated three types of solid-phase extraction columns, namely Waters Oasis HLB, Waters Oasis WCX, and Waters Oasis MCX. Preliminary experimental results showed that SCs could not be retained on weak cation-exchange mixed columns and strong cation-exchange mixed columns. Therefore, Waters Oasis WCX and Waters Oasis MCX are not suitable for the pretreatment of synthetic cannabinoids. The Waters Oasis HLB column is a hydrophilic and lipophilic balanced reverse-phase adsorbent. It is ideal for all substrates and capable of removing 95% of common substrate interferers such as phospholipids, fatty salts, and proteins [28]. Moreover, the adsorbent does not require adjustment and balancing steps, and the recovery rate is high. Gundersen et al. [29] compared different solid-phase extraction columns to quantify SC metabolites in urine by UHPLC-QTOF-MS, and finally selected HLB solid-phase extraction columns, which presented a fast through-

put and provided a more convenient protocol to some degree. Based on the physical and chemical properties of 11 synthetic cannabinoids and our pre-experiment result as well as the reported study, the Waters Oasis HLB column was selected as a solid-phase extraction column for pretreatment optimization.

The effects of the acetonitrile ratio of loading solution on the extraction recovery of 11 SCs were investigated. It is inferred that the use of ultrapure water as the loading liquid would result in the incomplete transfer of SCs to the SPE column since most SCs are lipophilic and insoluble in water. A certain volume of acetonitrile was added to the sample solution to make the SCs dissolve better in the loading solution. However, a large proportion of acetonitrile in the loading solution could lead to the elution of the tested substance during loading. Therefore, it is essential to select the appropriate ratio of loading solution. We kept the volume of the measured substance and rat urine constant and changed the volume of acetonitrile and ultrapure water. We set the volume of acetonitrile as 400, 600, and 800 μL. Certain volumes of ultrapure water were added to keep the total volume of 2 mL for the final sample solution. The results showed that the proportion of acetonitrile in the loading solution significantly affected the extraction recovery of 11 SCs (Figure 2). The extraction recovery rates were 52.7–122.3%, 85.3–119.5%, and 85.1–106.4% when the volumes of acetonitrile were 400, 600, and 800 μL, respectively. Similar results were obtained with the addition of 600 and 800 μL of acetonitrile, and with the principle of minimizing the amount of solvent used, 600 μL was finally chosen. Akamatsu et al. [30] also optimized the volume of acetonitrile to improve the separation effect of micellar electrokinetic chromatography-triple quadrupole mass spectrometry (MEKC-MS/MS) for the simultaneous determination of SCs and concluded that 20% *v/v* acetonitrile/water could produce better peak resolution, which implied that the proportion of acetonitrile significantly impacted the detection.

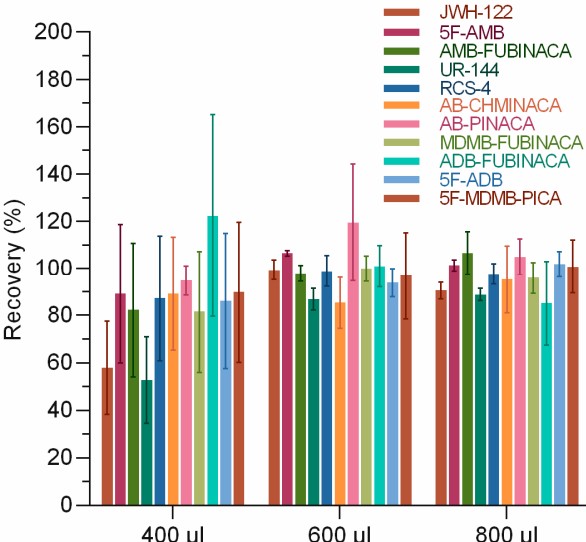

**Figure 2.** Optimization of acetonitrile volume in solid-phase extraction.

The effects of elution reagent volume on the extraction recovery of 11 SCs were investigated. The volume of elution reagent was set as 2, 3, 4, 5, and 6 mL to completely elute 11 SCs while reducing the amount of organic reagent. Theoretically, the recovery rate would gradually increase with the increase in elution reagent volume. Our results showed that the recovery rates of 11 SCs were 15.5–20.2%, 50.4–79.5%, 79.6–104%, 81.4–103.1%, and 82.7–107.9% when the volumes of elution reagents were 2, 3, 4, 5, and 6 mL, respectively (Figure 3). The results showed that the volume of the elution reagent greatly influenced the extraction recovery of SCs in lower amounts (2, 3, and 4 mL). In contrast, the recovery rate varied little when the elution reagent volume ranged from 4 to 6 mL. A good extraction recovery could be achieved for all 11 SCs when the volume of the elution reagent was 4 mL

(Figure 3). Borova et al. [31] also studied SPE conditions to detect ten new psychoactive substances and found that adding 4 mL methanol and then 8 mL of a mixture of MeOH/EtAc 1:1 had a better effect during the elution of SPE with a VARIAN (Vac Elut SPS 24) manifold and PolyClean 2H (200 mg) cartridges. The variation from our experimental results was due to the selection of SPE columns.

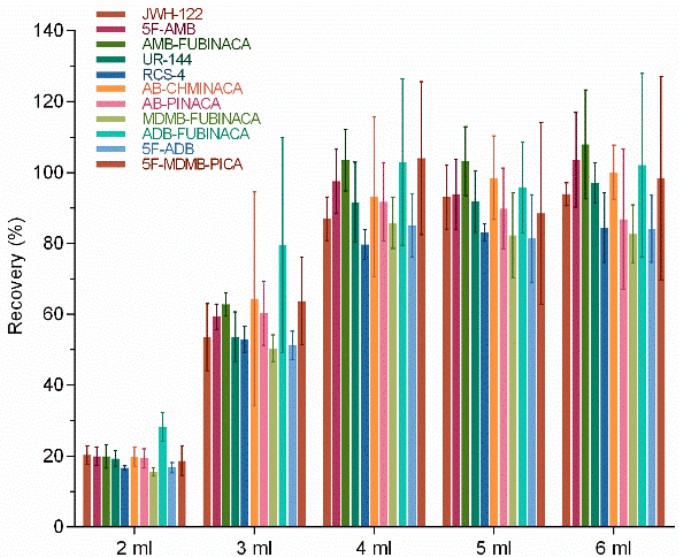

**Figure 3.** Effect of elution reagent volume on extraction recovery.

### 3.2. Establishment of Standard Curves

After pretreatment, 11 SC standard compounds were added into blank rat urine to configure matrix standard curves. The concentrations of 11 standard curves were 0.05, 0.1, 0.2, 0.5, 1, 5, 10, and 20 ng/mL. The matrix standard curve and the prepared samples were analyzed by LC-MS/MS. Linear regression was performed with the peak area as the ordinate coordinate and the standard solution concentration as the abscissa. The linear regression equation was obtained with a linear range of 0.1–500 ng/mL (Table 2). The results showed that all 11 target analytes were linear in the range of 0.1–500 ng/mL, and the coefficient of determination was between 0.993 and 0.999. The linear fit complex coefficient of determination in the linear regression equation is greater than 0.99, indicating good linearity [32].

**Table 2.** Synthetic cannabinoid urine matrix standard curve equations and limits of quantification.

| Compound | Urine Matrix Standard Curve | $R^2$ | LOQ (ng/mL) |
|---|---|---|---|
| JWH-122 | $y = 3.02938 \times 10^6 x + 5.09168 \times 10^4$ | 0.99841 | 0.01 |
| 5F-AMB | $y = 2.88293 \times 10^6 x + 5.16729 \times 10^4$ | 0.99835 | 0.03 |
| AMB-FUBINACA | $y = 2.86395 \times 10^6 x + 4.24124 \times 10^4$ | 0.99930 | 0.01 |
| UR-144 | $y = 1.48598 \times 10^6 x + 27734.81100$ | 0.99912 | 0.03 |
| RCS-4 | $y = 3.80986 \times 10^6 x + 1.12702 \times 10^5$ | 0.99578 | 0.03 |
| AB-CHMINACA | $y = 3.45358 \times 10^5 x + 5.58886 \times 10^4$ | 0.99829 | 0.1 |
| AB-PINACA | $y = 3.47936 \times 10^5 x + 20569.85965$ | 0.99894 | 0.1 |
| MDMB-FUBINACA | $y = 1.44915 \times 10^6 x + 5.04729 \times 10^4$ | 0.99726 | 0.03 |
| 5F-ADB | $y = 9.51164 \times 10^5 x + 6.20371 \times 10^4$ | 0.99592 | 0.03 |
| ADB-FUBINACA | $y = 2.35373 \times 10^6 x + 6.78715 \times 10^4$ | 0.99253 | 0.1 |
| 5F-MDMB-PICA | $y = 9.01193 \times 10^5 x + 8.52260 \times 10^4$ | 0.99541 | 0.1 |

### 3.3. Method Validation

Blank rat urine samples were treated with the solid-phase extraction method, and then mixed standard substances of 11 SCs were added to prepare 0.1, 1, and 10 ng/mL spiked rat urine sample solutions. When diluting urine with acetonitrile reagent, other substances coexist along with the analyte, including salts, amines, fatty acids, and other small molecules. The flow of these substances and analytes out of the spray needle can affect the atomization, volatilization, splitting, chemical reaction, and charging process of the analyte. This phenomenon caused a decrease or increase in ions entering the mass spectrometry, thereby affecting the reliability and accuracy of quantitative results. The final matrix effect range in our experiment was 76.7–106.1% (Table 3). In addition to AB-PINACA, the matrix effects of the 11 SCs are all within 80–100%, indicating that the influence of the matrix is small [33]. It can be concluded that the optimized solid-phase extraction method can effectively reduce the matrix effect.

**Table 3.** Synthetic cannabinoid matrix effects in urine matrix.

| Compound | 0.1 ng/mL Sample Matrix Effect% | 1 ng/mL Sample Matrix Effect% | 10 ng/mL Sample Matrix Effect% |
|---|---|---|---|
| JWH-122 | 99.3 | 94.6 | 97.8 |
| 5F-AMB | 104.2 | 106.1 | 97.7 |
| AMB-FUBINACA | 91.4 | 95.8 | 90.7 |
| UR-144 | 97.9 | 98.9 | 92.9 |
| RCS-4 | 83.6 | 91.9 | 101.8 |
| AB-CHMINACA | 89.8 | 93.8 | 95.3 |
| AB-PINACA | 76.7 | 96.8 | 94.0 |
| MDMB-FUBINACA | 98.0 | 81.6 | 92.6 |
| 5F-ADB | 92.5 | 91.9 | 98.1 |
| ADB-FUBINACA | 94.0 | 97.0 | 101.2 |
| 5F-MDMB-PICA | 105.2 | 92.0 | 97.9 |

The blank rat urine was mixed with 11 SCs with concentrations of 1, 10, and 100 ng/mL. Urine samples from rats (n = 6) were treated with SPE to evaluate the recovery and precision of target analytes extracted by SPE pretreatment. The spiked recoveries of 11 SCs at three levels were 69.90–118.39% (Table 4). As for the accuracy in method verification, the generally acceptable recovery rate ranges from 80 to 120%, indicating that the recovery rate of this method is within the acceptable range [34]. While some substances may have recovery rates below 80%, this method has undergone precision and accuracy testing, showing high levels of repeatability and accuracy. The intraday precision was 0.52–18.95% and indicated a good reproducibility of the method. The LOQ was the concentration of the analyte when the signal–noise ratio was 10. LOQs were 0.01 ng/mL for JWH-122 and AMB-FUBINACA, 0.03 ng/mL for 5F-AMB, UR-144, RCS-4, and 5F-ADB, and 0.1 ng/mL for AB-CHMINACA, AB-PINACA, ADB-FUBINACA, MDMB-FUBINACA, and 5F-MDMB-PICA (Table 2). This method meets the requirements of trace detection. The LOQs are lower than those in some previous studies that detected SCs in urine [26,35,36].

The constant evolution of SCs available to the public makes it nearly impossible for analytical laboratories to detect and identify the myriad of emerging compounds. After a new compound enters the market, it often takes months for these laboratories to develop the tools and methodology to test for the compound. This process is made more difficult by the delayed availability of certified reference standards. This complex and extensively variable chemistry is what contributes to the continual appearance of new substances on the market. The analytical challenges of keeping current with synthetic cannabinoids include the diversity of compounds and chemistries in the drug class, the large number of potential analogs and configurations, delays in obtaining analytical standards for addition to toxicological mass spectral databases, and having a consistent and universally agreed-upon system for nomenclature. This typically means that by the time the standard is available and added to an analytical scope, months of drug-positive cases could have been

missed and the synthetic cannabinoid in question could be no longer used within the drug-using populations of interest. Additionally, validation procedures and comprehensive metabolic studies need to be completed to identify the most appropriate metabolites for forensic analysis. Compounds may go undetected and significant numbers of forensic samples may be reported as negative before the laboratory realizes a new compound is being used. Future studies are needed to explore the simultaneous detection of the metabolites of 11 SCs.

**Table 4.** Spiked recoveries for 11 synthetic cannabinoid urine samples.

| Compound | 1 ng/mL | | 10 ng/mL | | 100 ng/mL | |
|---|---|---|---|---|---|---|
| | Recovery% | RSD% | Recovery% | RSD% | Recovery% | RSD% |
| JWH-122 | 78.85 | 2.14 | 75.04 | 0.52 | 79.63 | 0.85 |
| 5F-AMB | 76.03 | 18.68 | 75.84 | 0.92 | 95.85 | 0.59 |
| AMB-FUBINACA | 107.70 | 18.95 | 79.26 | 3.62 | 71.03 | 2.21 |
| UR-144 | 75.58 | 16.51 | 72.08 | 7.39 | 81.99 | 12.59 |
| RCS-4 | 85.94 | 9.77 | 78.78 | 5.31 | 76.26 | 7.89 |
| AB-CHMINACA | 93.76 | 11.43 | 87.90 | 15.67 | 98.82 | 10.16 |
| AB-PINACA | 118.39 | 9.98 | 69.90 | 4.38 | 82.00 | 7.29 |
| MDMB-FUBINACA | 73.07 | 3.97 | 76.04 | 6.62 | 91.38 | 6.03 |
| 5F-ADB | 94.59 | 10.25 | 74.59 | 3.30 | 92.04 | 7.02 |
| ADB-FUBINACA | 106.17 | 8.95 | 72.3 | 8.95 | 100.49 | 13.26 |
| 5F-MDMB-PICA | 82.75 | 10.22 | 70.38 | 6.32 | 74.61 | 5.44 |

## 4. Conclusions

In our study, the solid-phase extraction LC-MS/MS method was established to determine 11 synthetic cannabinoids in rat urine. When SPE conditions were optimized, the Waters Oasis HLB column had an excellent barrier effect on impurities in the biological samples. The acetonitrile ratio and the elution reagent volume were optimized to improve the recovery rate of tested samples. The results showed that the method had high sensitivity and good reproducibility, and the LOQ was 0.01–0.1 ng/mL less than that reported in the majority of literature. The spiked recoveries of 11 SCs were 69.90–118.39%, and the intraday precision was 0.52–18.95%. This method has been successfully applied to the analysis of cannabinoids in rat urine samples. The method is rapid and sensitive, and the preparation procedure is not complicated and requires only 0.4 mL of urine, which is of great significance for the rapid detection of drug-related cases and the strict prevention of drug-related criminal activities. The outlined method has proven to be accurate, precise, and free from matrix or chemical interferences. The implementation of this method in forensic testing can provide significant information for toxicologists. Future work will expand the assay analyte components as new compounds are reported in the marketplace.

**Supplementary Materials:** The following supporting information can be downloaded at: https://www.mdpi.com/article/10.3390/separations10030203/s1, Figure S1: The chemical structures of the 11 SCs.

**Author Contributions:** Conceptualization, J.W. and X.K.; methodology, F.Z.; validation, F.Z., X.K., J.W. and Y.F.; formal analysis, X.W. and L.Z.; resources, Y.F. and J.W.; data curation, F.Z., J.W., W.J. and X.W.; writing—original draft preparation, J.W. and F.Z.; writing—review and editing, J.Z.; visualization, F.Z., W.J., X.W. and J.Z.; supervision, J.Z. and Y.F.; project administration, Y.F.; funding acquisition, Y.F. and J.W. All authors have read and agreed to the published version of the manuscript.

**Funding:** This research was funded by Zhejiang Provincial Natural Science Foundation of China, grant number LTGC23B050003, National Key Research and Development Program of China, No. 2017YFC0803606.

**Institutional Review Board Statement:** The animal study protocol was approved by the Institutional Animal Care and Use Committee (IACUC), ZJCLA (protocol code ZJCLA-IACUC-20100005 and date of approval 10 October 2019).



**Informed Consent Statement:** Not applicable.

**Data Availability Statement:** Not applicable.

**Conflicts of Interest:** The authors declare no conflict of interest.

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
