# Peer review of "Rapid Simultaneous Determination of 11 Synthetic Cannabinoids in Urine by Liquid Chromatography-Triple Quadrupole Mass Spectrometry"

_separations, doi:10.3390/separations10030203_

Round 1

Reviewer 1 Report

This is a good work that evaluates a validated liquid chromatography tandem mass spectrometry method to accurately identify and quantitate 11 synthetic cannabinoid metabolites in urine sample. I have examined the submitted paper very carefully and I recommend its publication after some minor considerations. In the following, you can find my specific observations.

1. Page 3, line 124: Instead of “-20°C”, should be “-20 °C”

2. Page 4, line 163: Instead of "one/ten ng/mL", should be "1.0/10.0 ng/mL"

3. It woul be good to have the chemical structures of the SCs as a supplementary material, to have more information for the ms/ms análisis. 

Author Response

  1. Page 3, line 124: Instead of “-20°C”, should be “-20 °C”

RESPONSE: We appreciate the reviewer’s suggestion and we have revised “-20°C” to “-20 °C”. Please see the revised sentence, which was marked in red color.

Page 3, line 149-151

Stock standard solutions of 11 synthetic cannabinoids were prepared by dissolving 1 mg of analytes in 1 mL of acetonitrile to obtain the concentration of 1 mg/mL and were kept at -20 ℃.

  1. Page 4, line 163: Instead of "one/ten ng/mL", should be "1.0/10.0 ng/mL"

RESPONSE: We appreciate the reviewer’s suggestion and we have revised "one/ten ng/mL" to "1.0/10.0 ng/mL". Please see the revised sentence, which was marked in red color.

Page 5, line 189-191

Blank rat urine samples were treated with solid-phase extraction method, and then mixed standard substances of 11 SCs were added to prepare 0.1 ng/mL, 1.0 ng/mL, and 10.0 ng/mL spiked rat urine sample solution to calculate matrix effect.

  1. It would be good to have the chemical structures of the SCs as a supplementary material, to have more information for the ms/ms análisis.

RESPONSE: We appreciate the reviewer’s suggestion. In our revised manuscript, we have provided chemical structures of the SCs as a supplementary material.

Reviewer 2 Report

The manuscript by J. Wu et al. is focused on determination of eleven synthetic cannabinoids in spiked rat urine by LC-MS/MS. The authors also focused on sample pretreatment in the work. The manuscript could be of interest of many researchers but it needs revision before possible publication.

 Comments:

1    Page 2, lines 70-73: Supercritical fluid chromatography (SFC) is nowadays also used for synthetic and natural cannabinoids analysis and determination (e.g. Drug Test. Analysis 2018, 10, 222-228; Anal. Methods 2015, 7, 6056-6059; J. Chromatogr. B 2018, 1092, 332-342). In my opinion, the authors should add this method to the text with appropriate references.

2    Could the authors add a comment why they have chosen these eleven SCs?

3    Page 2, line 81: The authors mentioned that LC-MS is mainly used to analyze compounds with…”strong polarity”, however the compounds of interests are lipophilic as mentioned many times in the text.

4    Page 3, lines 123-125: The authors dissolved the SCs in methanol and diluted with acetonitrile. Please, add any explanation or comment on this.

5    It could be useful to add chemical structures of SCs used in the work.

6    It is not clear how the authors determined linear range 0.05 – 10 ng/mL while LOQ for four SCs is 0.1 ng/mL (higher than first calibration point (page 7)).

7    R2 is not correlation coefficient. It is coefficient of determination.

8    Could the authors specify how they calculated recovery for 100 ng/mL? This concentration is not included in “linear range”.

9    Page 9, lines 268-271: The authors obtained spiked recoveries in the range 69.90-118.39%. The generally acceptable recovery rate ranges from 80 to 120%. How the authors can conclude that the method is within the acceptable range if they are 10% below lower limit 80%?

 Minor comments:

1    Page 1, line 35 – “benzoyl indoles” are mentioned twice in this line

2    Page 2, lines 49 and 57: “Aleksandra and Davide” are first names

3    Page 2, line 61: better to use simultaneous instead of synchronous

Author Response

  1. Page 2, lines 70-73: Supercritical fluid chromatography (SFC) is nowadays also used for synthetic and natural cannabinoids analysis and determination (e.g. Drug Test. Analysis 2018, 10, 222-228; Anal. Methods 2015, 7, 6056-6059; J. Chromatogr. B 2018, 1092, 332-342). In my opinion, the authors should add this method to the text with appropriate references.

RESPONSE: We appreciate the reviewer’s suggestion. We have added this method to the revised manuscript.

Page 2, line 72-76

In recent years, the detection and analysis methods of SCs mainly include thin layer chromatography (TIC) [17], nuclear magnetic resonance (NMR) [18], infrared spectroscopy (IR)[19], gas chromatography-mass spectrometry (GC-MS) [20], liquid chromatography-mass spectrometry (LC-MS) [21], and supercritical fluid chromatography (SFC) [22].

  1. Could the authors add a comment why they have chosen these eleven SCs?

RESPONSE: We appreciate the reviewer’s suggestion. We have added a comment in revised manuscript.

Page 2, line 90-110

We selected 11 highly potent synthetic cannabinoids for our study, which may lead to many adverse reactions, including both conventional and novel ones. The selected compounds are presented in supplementary materials. N-(1-Amino-3-methyl-1-oxobutan-2-yl)-1-pe-ntyl-1H-indazole-3-carboxamide (AB-PINACA), N-(1-Amino-3-methyl-1-oxobutan-2-yl)-1-(cyclohexylmethyl)-1H-indazole-3-carboxamide (AB-CHMINACA), and N-(1-Methoxy-3,3-dimethyl-1-oxobutan-2-yl)-1-(4-fluorobenzyl)-1H-indazole-3-carboxamide (MDMB-FUBINACA) share a similar structure, containing N-alkyl imine and 1H-indole moieties. Their common feature is their extremely potent pharmacological effects and toxicity, which led to their classification as controlled substances. (1-Pentyl-1H-indol-3-yl)(4-methylnaphthalen-1-yl)methanone (JWH-122), 1-Pentyl-3-(4-methoxybenzoyl)indole (RCS-4), and methyl 2-[1-(5-fluoropentyl)-1H-indazole-3-carboxamido]-3,3-dimethylbutanoate (5F-ADB) also have similar structures, containing 1H-indole and arylketone moieties. These compounds are highly toxic and classified as controlled substances as well. ADB-FUBINACA has a structure similar to that of AB-PINACA, AB-CHMINACA, and MDMB-FUBINACA, containing N-alkyl imine and 1H-indole moieties, but with even greater toxicity. 5F-MDMB-PICA, 5F-AMB, UR-144, and AMB-FUBINACA have similar structures, containing indole moieties and alkyl imine, but with different substituent groups that result in different pharmacological effects and toxicity. Overall, synthetic cannabinoids are characterized by their structural diversity, high toxicity, and complex pharmacological effects. Due to their potential for abuse and danger, effective monitoring and control of these substances is necessary.

  1. Page 2, line 81: The authors mentioned that LC-MS is mainly used to analyze compounds with…”strong polarity”, however the compounds of interests are lipophilic as mentioned many times in the text.

RESPONSE: We appreciate the reviewer’s suggestion. In our revised manuscript, we have provided the explanation in the revised manuscript as follows.

Page 2, line 84-86

LC-MS is primarily employed for the analysis of compounds with high thermal instability, strong polarity, lipophilicity, or high molecular weight, with the advantages of strong separation ability, low detection limit, high sensitivity, and high accuracy.

  1. Page 3, lines 123-125: The authors dissolved the SCs in methanol and diluted with acetonitrile. Please, add any explanation or comment on this.

RESPONSE: We appreciate the reviewer’s concern. We have corrected the chemical name of reagent and provided the explanation in the revised manuscript as follows.

Page 3, line 149-151

Stock standard solutions of 11 synthetic cannabinoids were prepared by dissolving 1 mg of analytes in 1 mL of acetonitrile to obtain the concentration of 1 mg/mL and were kept at -20 ℃.

  1. It could be useful to add chemical structures of SCs used in the work.

RESPONSE: We appreciate the reviewer’s concern. We have added chemical structures of SCs as supplementary material.

  1. It is not clear how the authors determined linear range 0.05 – 10 ng/mL while LOQ for four SCs is 0.1 ng/mL (higher than first calibration point (page 7)).

RESPONSE: We appreciate the reviewer’s concern. We have corrected the linear range and provided the explanation in the revised manuscript as follows.

Page 7, line 270-273

The linear regression equation was obtained with a linear range of 0.1-500 ng/mL (Table 2). The results showed that all 11 target analytes were linear in the range of 0.1-500 ng/mL, and the correlation coefficient was between 0.993-0.999.

  1. R2 is not correlation coefficient. It is coefficient of determination.

RESPONSE: We appreciate the reviewer’s concern. We have corrected the term and provided the explanation in the revised manuscript as follows.

Page 1, line19-20

The results showed that the linear coefficients of determination of 11 types of synthetic cannabinoids ranged from 0.993 to 0.999, the limit of quantitation ranged from 0.01 to 0.1 ng/mL, and the spiked recoveries ranged from 69.90% to 118.39%.

Page 7, line 270-275

The linear regression equation was obtained with a linear range of 0.1-500 ng/mL (Table 2). The results showed that all 11 target analytes were linear in the range of 0.1-500 ng/mL, and the coefficient of determination was between 0.993-0.999. The linear fit complex coefficient of determination in the linear regression equation is greater than 0.99, indicating good linearity [32].

  1. Could the authors specify how they calculated recovery for 100 ng/mL? This concentration is not included in “linear range”.

RESPONSE: We appreciate the reviewer’s concern. We have corrected the linear range and provided the explanation in the revised manuscript as follows.

Page 7, line 270-273

The linear regression equation was obtained with a linear range of 0.1-500 ng/mL (Table 2). The results showed that all 11 target analytes were linear in the range of 0 ng/mL, and the correlation coefficient was between 0.993-0.999.

  1. Page 9, lines 268-271: The authors obtained spiked recoveries in the range 69.90-118.39%. The generally acceptable recovery rate ranges from 80 to 120%. How the authors can conclude that the method is within the acceptable range if they are 10% below lower limit 80%?

RESPONSE: We appreciate the reviewer’s concern. We have provided the explanation in the revised manuscript as follows.

Page 9, line 298-300

While some substances may have recovery rates below 80%, this method has under-gone precision and accuracy testing, showing high levels of repeatability and accuracy.

Minor comments:

  1. Page 1, line 35 – “benzoyl indoles” are mentioned twice in this line

RESPONSE: We appreciate the reviewer’s concern. We have improved the explanation in the revised manuscript as follows.

Page 1, line 34-36

SCs belong to psychoactive substances, including benzoyl indoles, naphthyl indoles, cyclohexyl phenols, naphthyl pyrrole, and diamond indoles [3].

  1. Page 2, lines 49 and 57: “Aleksandra and Davide” are first names

RESPONSE: We appreciate the reviewer’s concern. We have corrected the names of the authors in the cited references.

Page 2, line 49-59

Ellert-Miklaszewska et al. found that SCs could activate the killing pathway of human glioma cells and had anti-tumor activity [7]. However, more negative effects were mentioned. Ingestion of cannabinoids could cause many diseases, as appeared and de-scribed in the literature, including psychosis [8], intoxication [9], tachycardia [10], changes in blood pressure [11], and atrial fibrillation [12]. Some reports declared fatality directly related to the toxic effects of synthetic cannabinoid [13, 14]. Tomiyama et al. reported that SCs induced cell apoptosis by regulating the cascade of cystatin, which stressed that the abuse of cannabinoids could lead to neurological brain problems [15]. Radaelli et al. concluded that SCs could activate CB1 receptors in cardiomyocytes, participate in the production of reactive oxygen species, ATP consumption, and cell death, and then induce cardiotoxicity [16].

  1. Page 2, line 61: better to use simultaneous instead of synchronous

RESPONSE: We appreciate the reviewer’s concern. We have corrected the “synchronous” to “simultaneous” in the revised manuscript as follows.

Page 2, line 63-64

Due to new SCs constantly emerging in the market, there is an urgent need to up-date the simultaneous detection method of SCs.

Round 2

Reviewer 2 Report

The authors corrected the manuscript according to the comments. In my opinion, the overall quality of the manuscript increased. I do not have any additional comments.